# A Conversation with ChatGPT about Digital Leadership and Technology Integration: Comparative Analysis Based on Human–AI Collaboration

Turgut Karakose [1], Murat Demirkol [2], Ramazan Yirci [3,*], Hakan Polat [2], Tuncay Yavuz Ozdemir [2] and Tijen Tülübaş [1]

1 Faculty of Education, Kutahya Dumlupınar University, 43100 Kütahya, Türkiye; turgut.karakose@dpu.edu.tr (T.K.); tijen.tulubas@dpu.edu.tr (T.T.)
2 Faculty of Education, Firat University, 23119 Elazığ, Türkiye; mdemirkol@firat.edu.tr (M.D.); hakanpolat@firat.edu.tr (H.P.); tyozdemir@firat.edu.tr (T.Y.O.)
3 Faculty of Education, Sutcuimam University, 46050 Kahramanmaras, Türkiye
* Correspondence: yirci@ksu.edu.tr

**Abstract:** Artificial intelligence (AI) is one of the ground-breaking innovations of the 21st century that has accelerated the digitalization of societies. ChatGPT is a newer form of AI-based large language model that can generate complex texts that are almost indistinguishable from human-generated text. It has already garnered substantial interest from people due to its potential utility in a variety of contexts. The current study was conducted to evaluate the utility of ChatGPT in generating accurate, clear, concise, and unbiased information that could support a scientific research process. To achieve this purpose, we initiated queries on both versions of ChatGPT regarding digital school leadership and teachers' technology integration, two significant topics currently discussed in educational literature, under four categories: (1) the definition of digital leadership, (2) the digital leadership skills of school principals, (3) the factors affecting teachers' technology integration, and (4) the impact of digital leadership on teachers' technology integration. Next, we performed a comparative analysis of the responses generated by ChatGPT-3.5 and ChatGPT-4. The results showed that both versions were capable of providing satisfactory information compatible with the relevant literature. However, ChatGPT-4 provided more comprehensive and categorical information as compared to ChatGPT-3.5, which produced responses that were more superficial and short-cut. Although the results are promising in aiding the research process with AI-based technologies, we should also caution that, in their current form, these tools are still in their infancy, and there is a long way to go before they become fully capable of supporting scientific work. Meanwhile, it is significant that researchers continue to develop the relevant knowledge base to support the responsible, safe, and ethical integration of these technologies into the process of scientific knowledge creation, as Pandora's box has already been opened, releasing newer opportunities and risks to be tackled.

**Keywords:** digital leadership; technology integration; school leadership; ChatGPT; artificial intelligence; generative AI; AI in education; chatbot

## 1. Introduction

People have witnessed numerous ground-breaking technologies that have altered fundamental operations in their lives, particularly since the beginning of the 21st century. However, recent developments in machine learning and artificial intelligence are establishing the ground for unprecedented breakthroughs that will revolutionize the way we live.

Artificial intelligence is defined as the ability of machines to produce automated solutions to accomplish the tasks assigned to them, particularly with minimum or even without human intervention (Aghion et al. 2018; Blumenthal 2017; McCarthy et al. 2006;

Shubhendu and Vijay 2013). It is, in fact, a learning system that uses mathematical algorithms to perform tasks that require human intelligence (de Saint Laurent 2018; Stokes and Palmer 2020) and incorporates the use of other systems such as fuzzy logic, intelligent factors, genetic algorithm, artificial neural networks, and expert systems (Elmas 2021; Öngöz 2020). Machine learning, a sub-discipline of supervised and unsupervised learning fields, is a significant component of building artificial intelligence systems, if not the only one (Russel and Norvig 1995), and plays a significant role in optimizing the performance of artificial intelligence by training it on sample data or past experience (Alpaydin 2004). In other words, machine learning enables analyzing patterns in data in order to make predictions and suggestions based on their self-created algorithms (Mitchell 1997).

Artificial intelligence technologies such as robotic systems, autonomous vehicles, facial recognition, natural language processing, and virtual agents are currently used to solve a wide variety of problems (Berente et al. 2021) and play an important role in the digitalization of societies (Cooper 2023). ChatGPT (Chat Generative Pre-trained Transformer) is a new form of generative artificial intelligence that is capable of generating texts with human-like language (OpenAI 2023a). Artificial intelligence tools such as ChatGPT are designed to generate complex texts that are indistinguishable from human-generated text, and they can be utilized in a wide variety of contexts (Dwivedi et al. 2023). With the use of machine learning algorithms, ChatGPT was trained on the structure of the language by processing terabytes of data (OpenAI 2022; Scharth 2022) so that it can present meaningful content upon users' requests (Halaweh 2023). After its launch in November 2022 by OpenAI Limited Partnership, ChatGPT has rapidly reached more than 100 million users (Milmo 2023), and people have tested its utility for a wide variety of use cases such as wiring poems and stories, taking life advice, producing software codes, performing mathematical calculations and statistical analysis as well as translating texts into different languages (Atlas 2023; D'Amico et al. 2023; Karakose 2023; Mhlanga 2023; Scharth 2022; van Dis et al. 2023).

After the release of ChatGPT version 3.5, Open AI continued its work on ChatGPT to equip it with better capabilities while reducing the limitations of the previous versions. On 15 March 2023, they opened the latest version, i.e., Chat-GPT-4, for public use. As explained in its technical report (OpenAI 2023b), the performance of ChatGPT-4 was much beyond that of ChatGPT-3.5 on many tasks. For example, ChatGPT-4 not only outperformed ChatGPT-3.5 but also most human test takers in various exams originally designed for humans. ChatGPT-4 was also proven to perform better than any other state-of-the-art-systems available at the time of its release, and it demonstrated superior performance in differentiating fact from incorrect information, following user intent, reasoning, and generating concise responses. This may have particularly resulted from the fact that ChatGPT-3.5 was trained on 175 billion parameters, while GPT 4 was trained on 100 trillion parameters (Zaitsu and Jin 2023). The developments also helped reduce its likelihood of generating hallucinated or incorrect information, mentioned as a serious limitation of ChatGPT-3.5 (OpenAI 2023b). Recent research conducted after the release of ChatGPT-4 (e.g., Teebagy et al. 2023; West 2023; Tülübaş et al. 2023) also indicates that ChatGPT-4 demonstrates capabilities far beyond its previous version, even in showing emotions (Zhao et al. 2023), performing inductive reasoning and inferring peoples' feelings or perspectives (Michail et al. 2023), despite acknowledging that there is still much room for improvement if it is expected to act like a human.

ChatGPT can continuously improve itself by using the feedback received from its users (Farrokhnia et al. 2023; Mann 2023), and has the ability to find and summarize information retrieved from the internet literature in response to users' queries (Cascella et al. 2023). Hence, ChatGPT is capable of performing complex tasks such as assisting language learning (Jia et al. 2022), promoting critical thinking (Hapsari and Wu 2022), improving academic writing skills (Zhai 2022), and developing programming skills (Biswas 2023). It is even considered to be a useful tool for reducing teachers' workload in designing and performing instruction (Qadir 2023). ChatGPT can also provide significant guidance at different stages

of a research process, such as brainstorming, data collection and analysis, critical evaluation of data, and reporting research results in written form.

In addition to the above-mentioned benefits, ChatGPT also has some shortcomings and weaknesses. Some scholars state that responses generated by ChatGPT do not have philosophical depth (Bogost 2022; Gao et al. 2023), and it could produce answers irrelevant to the topic of the query (Gupta et al. 2023). Another significant concern is about the reliability and accuracy of the responses, which are generated based on the corpus data on which it is trained (Lecler et al. 2023; Sallam 2023; Stokel-Walker and Van Noorden 2023). Therefore, ChatGPT has the potential to produce biased responses since it cannot identify the bias existing in its training corpus or the internet resource it reaches (Barocas and Selbst 2016; Zhai 2022). In addition, it is known that ChatGPT can produce inaccurate or irrelevant references when asked to provide the source of reference for the information generated (Choi et al. 2023), and could cause plagiarism and copyright problems as some responses could be directly taken from published research (Kasneci et al. 2023).

Although much research is conducted on the use of ChatGPT, particularly ChatGPT-3.5, with regard to its performance on several tasks (Sallam 2023; Taecharungroj 2023), a review of the literature reveals a limited amount of research on the use of ChatGPT to support scientific work. Considering this gap in the literature, the current study aims to conduct a research process in collaboration with human and artificial intelligence using the responses taken from ChatGPT-3.5 and ChatGPT-4 on a specific research topic in the field of educational administration. We particularly focused on the investigation of two recent and related topics in the field–digital leadership and teachers' technology integration. To achieve this purpose, we first had a conversation with the two versions of ChatGPT about school principals' digital leadership and teachers' technology integration. Next, we conducted a comparative analysis of their responses. Using both versions of ChatGPT, we aimed to conduct a comparative assessment of whether they could provide accurate, clear, concise, and unbiased information.

*Digital Leadership and Teachers' Technology Integration*

Digitalization has become an inevitable truth in the current age of technological breakthroughs. Digital technologies transform many operations in daily life, such as production, logistics, communication, and human resources management (Oberer and Erkollar 2018). As modern organizations transform into digital work environments, digital leadership has become significant in enabling successful digital transformation (Arham et al. 2023; Ghavifekr and Pei 2023). Digital leadership is used as an umbrella term that encompasses several leadership styles, such as technology leadership, virtual leadership, e-leadership, and leadership 4.0, all of which are used interchangeably in the literature (Karakose et al. 2022). Digital leadership also involves utilizing technology effectively so as to provide better work conditions for employees (Berisha-Gawlowski et al. 2021). In the context of education, Zhong (2017) defines digital leadership as using instructional technologies such as digital devices, services, and resources to inspire and lead the digital transformation of the school, create and maintain a culture of digital learning, support and develop technology-based professional development, and eventually enable a digital-friendly school environment. From this point of view, digital leadership can be defined as a technology-based leadership model that involves the effective use of technology to promote schools' functioning.

The use of advanced technologies in the educational context has recently become widely acknowledged (Al-Ruz and Khasawneh 2011), and teachers of all grades are now expected to integrate technology successfully into their classrooms (Keengwe et al. 2008). Teachers felt this need even more intensely during the COVID-19 pandemic, which has accelerated the transformation of education systems around the world by encouraging or even mandating the integration of digital technologies into education (Hamzah et al. 2021). Educational leaders and policy makers argue that digital technology integration has become the key to fostering student engagement and achievement (Howley et al. 2011). Scientists

have emphasized that technology integration improves students' critical and creative thinking skills, and increases their academic success and motivation (Siddiqui et al. 2020; Yilmaz 2021). As a result, it is frequently recommended to develop the digital skills of both teachers and students by integrating advanced technologies into previously human-centered practices (Ruggiero and Mong 2015).

The successful integration of technology into education entails teachers' effective use of digital technologies in their classroom practices. Therefore, the attitudes and beliefs of teachers towards using these technologies significantly affect technology integration (Alghamdi and Prestridge 2015; Christensen 2002; Ertmer et al. 2012; Kim et al. 2013; Liu 2011). In addition, contextual factors such as the availability of these technologies and having easy access to such resources also promote technology integration to a great extent (Howley et al. 2011; Sauers and McLeod 2018; Shuldman 2004). As stated by several scholars, providing teachers with administrative encouragement and support is significant in enabling the successful integration of new technologies into education (Al-Ruz and Khasawneh 2011; Chang 2012; Keengwe et al. 2008; Kim et al. 2013; McLeod 2015). McLeod (2015) emphasized that the technology-based transformation of schools necessitates instructional leadership with a good vision of technology integration. Therefore, school principals are considered to have a significant role in developing teachers' capabilities to conduct technology-integrated instruction and motivating them to use the latest technologies for the benefit of students (Chang 2012; Navaridas-Nalda et al. 2020).

In the face of rapid digital transformation, both teachers and school principals are expected to become proficient in using technology. According to Hensellek (2020), as digital leaders, school principals not only need to provide a strategic vision for a digital future but also have the necessary digital skills and attitudes to achieve this vision in collaboration with all the stakeholders. Aksal (2016) emphasized that digital leadership could enhance technology integration into both the teaching-learning processes and the management of schools. School principals who perform successful digital leadership can also create a digital learning culture in schools, which supports digital transformation and technology-based professional development (Karakose et al. 2021; Karakose and Tülübaş 2023). Therefore, school principals should be active participants and role models in technology integration and should espouse technology integration as the core task of their leadership (McLeod 2015) so as to enable the digital transformation of schools.

## 2. Materials and Methods

This study aims to conduct research with the collaboration of human and artificial intelligence in order to see the possible contribution of AI-based systems to generating accurate, clear, concise, and unbiased information. With this purpose, we conducted queries on digital school leadership and its potential impact on teachers' technology integration using two versions of an AI-based large language model (LLM): ChatGPT-3.5 and ChatGPT-4. We preferred to conduct a comparative assessment, which could help identify the similarities and differences of responses yielded by each version and observe whether ChatGPT-4 brings in any advances and innovations with regard to response generation.

We conducted the research in three stages. First, we reviewed the relevant literature about digital leadership and teachers' technology integration so as to develop our categories to interrogate ChatGPT. At this stage, we also prepared a list of questions to get responses for each category of query. During this initial stage, all the researchers first worked individually. Later, we performed a panel discussion to develop our final list of categories and related questions. Eventually, we agreed upon four categories with a single question to be used for each: (1) the definition of digital leadership, (2) the digital leadership skills of school principals, (3) the factors affecting teachers' technology integration, and (4) the impact of digital leadership on teachers' technology integration.

Second, we made simultaneous queries on ChatGPT-3.5 and ChatGPT-4 on 13 April 2023, using the question developed for each category without including any additional prompts or questions to minimize human interference and guidance during the queries.

After recording responses given by each version of ChatGPT, we moved on to the third stage: the analysis of responses yielded for each category. While analyzing the quality of these responses, we used two different rating systems. To evaluate the inter-rater agreement of researchers on the quality of responses, we calculated Cohen's kappa for each category. Cohen's kappa is often used in social and medical sciences as a chance-corrected means of assessing inter-rater agreement on a nominal scale. It is considered to be an efficient measurement to determine whether the degree of agreement is obtained for real or by chance (Sun 2011; Warrens 2015). Hence, Cohen's kappa is used as a robust statistic to assess inter-rater agreement, and the results are used to evaluate the quality of information assessed by two raters (Vieira et al. 2010). In the current study, we calculated the value of Cohen's kappa for each category using the package program SPSS version 26 and interpreted the results according to the benchmark suggested by Landis and Koch (1977): 0.00–0.20 = slight inter-rater agreement, 0.21–0.40 = fair agreement, 0.41–0.60 = moderate agreement, 0.61–0.80 = strong agreement, and 0.81–1.00 = almost perfect agreement.

To assess (1) the accuracy, (2) clarity, (3) conciseness, and (4) the potential for bias in the information provided by each version of ChatGPT, we developed a four-dimensional rating scale with three points of rating: (1) meant that the information is completely inaccurate, unclear, unconcise, or biased, (2) meant that it was partly accurate, clear, concise, or biased, and (3) meant that it was completely accurate, clear, concise, or unbiased. Using this rating scale, all researchers individually rated the responses yielded for each category in terms of accuracy, clarity, conciseness, and potential bias, and later the averages of these rating points were taken for each category of evaluation.

As data for the current study were tested based on the individual understandings and evaluations of six researchers who participated in the research, it would also be proper to declare some characteristics of the researchers to support the credibility of the results. All six researchers who conducted the analysis were educational specialists working on two distinct sub-fields. Three of the researchers were experts on educational administration, management, and leadership, while the other three worked particularly in the area of educational technologies and were experts on designing newer technologies to support education. In fact, all researchers were familiar with the topics under investigation and were interested in exploring technological innovations. The diversity of backgrounds was positive in that expertise in educational management and leadership helped evaluate the accuracy and conciseness of information with regard to digital leadership, while expertise in educational technologies helped evaluate the responses with regard to teachers' technology integration. It also enabled reflection of different views regarding the responses. However, it should also be noted that the personal characteristics, expectancies, and views of each researcher could also ignite some level of bias in their ratings, although this is difficult to uncover for sure. However, the results of Cohen's kappa and the inclusion of ratings from six researchers (rather than a single researcher) were useful in eliminating personal bias in the analysis.

## 3. Results

This section reports on the responses generated by ChatGPT-3.5 and ChatGPT-4 on the previously defined four categories of digital school leadership and its impact on teachers' technology integration: (1) the definition of digital leadership, (2) the digital leadership skills of school principals, (3) the factors affecting teachers' technology integration, and (4) the impact of digital leadership on teachers' technology integration. In particular, the section provides the comparative evaluation of responses given by two versions of ChatGPT for each category, the results of researcher ratings on the accuracy, clarity, conciseness, and bias potential of information provided by these responses, and the inter-rater agreements on the quality of information with this regard (i.e., the Cohen kappa values). We also included the snapshots of responses generated by ChatGPT-3.5 and ChatGPT-4 to illustrate the results for each category.

We started our interrogation with 'the definition of digital leadership in education', and sample excerpts from the responses of ChatGPT-3.5 and ChatGPT-4 are presented in Figure 1.

---

RESEARCH THEME: The Definition of Digital Leadership

---

### Model: GPT-3.5

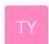 What is digital leadership in education?

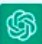 Digital leadership in education refers to the ability of educational leaders to effectively integrate technology into teaching and learning. It involves developing and implementing strategies that promote the use of digital tools and resources in education, and providing guidance and support to educators in the effective use of these tools.

Digital leadership encompasses a range of skills and competencies, including technological proficiency, strategic thinking, communication, collaboration, and change management. It involves staying up-to-date with emerging technologies and trends, and making informed decisions about which tools and platforms are best suited for specific educational contexts and learning objectives.

### Model: GPT-4

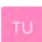 What is digital leadership in education?

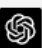 Digital leadership in education refers to the effective use of technology and digital tools by educational leaders, such as school administrators, principals, and teachers, to enhance teaching, learning, and overall school management. It encompasses various aspects of leadership, including vision, strategy, communication, and collaboration, all of which are geared toward creating a transformative and innovative learning environment. Key aspects of digital leadership in education include:

1. Vision and strategy: Digital leaders in education develop a clear vision and strategy for integrating technology into the curriculum and school culture. They create a roadmap for the successful implementation of digital tools, ensuring alignment with educational goals.
2. Professional development: Digital leaders prioritize professional development for themselves and their staff, staying current with new technologies, digital pedagogies, and best practices in the field. They promote and support continuous learning among

---

**Figure 1.** Sample excerpts from responses of ChatGPT-3.5 and ChatGPT-4 for the definition of digital leadership in education (generated on 13 April 2023).

ChatGPT-3.5 defined digital leadership as the ability of educational leaders to integrate technology into their learning and teaching processes, while ChatGPT-4 defined it as the ability to integrate technology into school management processes as well as learning and teaching processes, stating that school administrators, principals and teachers are educa-

tional leaders who can assume a digital leadership role in schools. In addition, ChatGPT-4 stated that digital leadership requires vision and strategy, digital citizenship, communication and collaboration, innovation and adaptability, and data-driven decision making, which would, in turn, help create a transformative and innovative learning environment.

Both versions of ChatGPT underlined similar dimensions of digital leadership, such as supporting and promoting the professional development of educators, creating a digital-friendly culture at school, and developing strategies to support this culture using innovative technologies, communicating and cooperating with all stakeholders, implementing effective decision-making processes, and enhancing technology used for the benefit of students. However, while ChatGPT-3.5 associated these skills with teachers and teaching-learning processes, ChatGPT-4 tended to associate them with the management roles of school administrators in addition to teaching and learning processes. In addition, ChatGPT-4 was able to generate more comprehensive, detailed, and categorized information as compared to ChatGPT-3.5.

The average values of ratings based on the four-dimensional rating scale for the accuracy, clarity, conciseness, and potential bias in the information provided by ChatGPT-3.5 were 2.67, 2.50, 2.16, and 2.83, respectively, while it was 2.83, 2.67, 2.83, and 2.83 for ChatGPT-4 responses (see Table 1). These results indicate that both versions of ChatGPT were able to generate satisfactory information on the definition of digital leadership in education, whereas ChatGPT-4 received better rates, particularly with regard to the accuracy and conciseness of information. For instance, as elaborated above, ChatGPT-4 underlined that educational leaders are not limited to school principals. Teachers and other staff with administrative roles could assume the role of digital leadership, while ChatGPT-3.5 did not make this clarification. Likewise, while presenting the roles of a digital leader, ChatGPT-4 presented these roles under categorical sub-headings such as vision and strategy, professional development, digital citizenship, communication and collaboration, innovation and adaptability, data-driven decision making, and offered detailed information about each of these categories in relation to digital leadership. ChatGPT-3.5, on the other hand, just summarized these roles in a single, short paragraph. Although both versions tend to offer a short summary of the information generated for each question, it was also evident that ChatGPT-4 represented a more positive stance towards adopting digital leadership (e.g., "*By embracing digital leadership . . .* "), as also underlined in the literature, while ChatGPT-3.5 just summarized the information it provided (e.g., "*Overall, digital leadership is about . . .* ").

**Table 1.** The Rating Scores for the Definition of Digital Leadership.

| | ChatGPT-3.5 | | | | | | | ChatGPT-4 | | | | | | |
| --- | --- | --- | --- | --- | --- | --- | --- | --- | --- | --- | --- | --- | --- | --- |
| | R1 | R2 | R3 | R4 | R5 | R6 | $\bar{x}$ | R1 | R2 | R3 | R4 | R5 | R6 | $\bar{x}$ |
| Accuracy | 2 | 3 | 3 | 2 | 3 | 3 | 2.67 | 3 | 3 | 3 | 2 | 3 | 3 | 2.83 |
| Clarity | 2 | 3 | 2 | 3 | 2 | 3 | 2.50 | 2 | 3 | 3 | 2 | 3 | 3 | 2.67 |
| Conciseness | 2 | 3 | 2 | 2 | 2 | 2 | 2.16 | 3 | 2 | 3 | 3 | 3 | 2 | 2.83 |
| Bias Possibility | 2 | 3 | 3 | 3 | 3 | 3 | 2.83 | 2 | 3 | 3 | 3 | 3 | 3 | 2.83 |

The Cohen's kappa value for the quality of responses under this category was calculated as 0.89 for ChatGPT-3.5 and 0.94 for ChatGPT-4. According to Landis and Koch's (1977) benchmark, these results indicate an 'almost perfect' agreement of inter-raters with regard to the quality of these responses in terms of accuracy, clarity, conciseness, and potential bias.

We continued our interrogation with the 'digital leadership skills of school principals', and sample excerpts from the responses of both versions of ChatGPT are shown in Figure 2.

The digital leadership skills listed by both versions of ChatGPT were school principals' proficiency in integrating technology into teaching and learning processes. The responses had some similar aspects as well as differences. Both versions underlined technology profi-

ciency/literacy, communication, collaboration, and change management as the required skills of digital leaders. However, unlike ChatGPT-3.5, ChatGPT-4 emphasized the role of school principals in driving digital transformation, focusing on newer concepts such as digital literacy and fostering digital citizenship.

RESEARCH THEME: Digital Leadership Skills of School Principals

Model: GPT-3.5

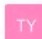 What are the school principals' digital leadership skills?

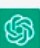 School principals' digital leadership skills include a range of competencies that enable them to effectively integrate technology into teaching and learning, and to lead their schools in a digital age. Some of these skills include:

1. Technology proficiency: School principals should have a good understanding of technology tools and resources that are relevant to their school's needs, and should be able to use these tools themselves.
2. Strategic thinking: Principals need to be able to develop and implement a strategic plan for the use of technology in their school, and to ensure that the use of technology is aligned with the school's goals and objectives.

Model: GPT-4

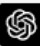 What are the school principals' digital leadership skills?

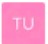 School principals play a crucial role in driving digital transformation in educational settings. As digital leaders, they must possess a range of skills to effectively integrate technology into their schools and promote innovative teaching and learning practices. Key digital leadership skills for school principals include:

1. Visionary leadership: Principals need to develop a clear vision for how technology can enhance teaching and learning and communicate this vision effectively to all stakeholders, including teachers, students, and parents. They must be able to create and implement strategic plans to achieve these goals.
2. Change management: School principals must be skilled in managing change, guiding their school community through the process of digital transformation. This includes fostering a culture that embraces innovation, addressing resistance to change, and providing ongoing support to ensure successful adoption of new technologies and practices.

**Figure 2.** Sample excerpts from responses of ChatGPT-3.5 and ChatGPT-4 for principals' digital leadership skills (generated on 13 April 2023).

In addition, ChatGPT-3.5 stated that strategic thinking and ethical leadership are among the digital leadership skills of school principals. ChatGPT-4, on the other hand, emphasized current concepts such as digital transformation, the digital age, and data-based

decision making, which are more frequently identified as digital leadership skills in the literature. In addition, ChatGPT-4 emphasized the responsibilities of school administrators on teachers, students, families, and other stakeholders by highlighting the direction, guidance, and encouragement aspects of school administration. Although both versions presented information under categorical sub-headings, the responses by ChatGPT-4 were clearly more detailed and thorough compared to those generated by ChatGPT-3.5 for this category of query. For example, ChatGPT-3.5 listed 'technology proficiency' as the first skill, stating that "*school principals should have a good understanding of technology tools and resources that are relevant to their school's needs, and should be able to use these tools themselves*". However, ChatGPT-4 mentioned the same skill as 'technological literacy', which is a more up-to-date terminology that comprises not only the knowledge and skills of technology use but also the ability to recognize its "*risks, challenges, and benefits*" to be able to "*make informed decisions about technology adoption, infrastructure, and resource allocation*". In the same vein, ChatGPT-4 lists 'digital citizenship' as a significant skill of a digital leader in education, defining digital citizenship as "ensuring that students and staff are aware of their rights and responsibilities in the digital world, promot[ing] responsible online behavior". Digital citizenship is one of the current topics of discussion in the field of education, and the fact that ChatGPT-4 underlined this aspect of digital leadership while ChatGPT-3.5 did not, implies that GPT-4 could generate more accurate and comprehensive information.

The average values of ratings based on the four-dimensional rating scale for the accuracy, clarity, conciseness, and potential bias in the information provided by ChatGPT-3.5, were 2.67, 2.67, 2.00, and 2.83, respectively, while it was 2.83, 2.83, 3, and 3 for ChatGPT-4 responses (see Table 2). These results indicate that both versions of ChatGPT were able to generate satisfactory information with regard to the digital leadership skills of school principals, while ChatGPT-4 received better rates in all dimensions of assessment.

**Table 2.** The Rating Scores for the Digital Leadership Skills.

| | ChatGPT-3.5 | | | | | | | ChatGPT-4 | | | | | | |
|---|---|---|---|---|---|---|---|---|---|---|---|---|---|---|
| | R1 | R2 | R3 | R4 | R5 | R6 | $\bar{x}$ | R1 | R2 | R3 | R4 | R5 | R6 | $\bar{x}$ |
| Accuracy | 3 | 2 | 2 | 3 | 3 | 3 | 2.67 | 3 | 2 | 3 | 3 | 3 | 3 | 2.83 |
| Clarity | 3 | 3 | 2 | 3 | 3 | 2 | 2.67 | 3 | 3 | 3 | 2 | 3 | 3 | 2.83 |
| Conciseness | 2 | 2 | 2 | 2 | 2 | 2 | 2.00 | 3 | 3 | 3 | 3 | 3 | 3 | 3.00 |
| Bias Possibility | 2 | 3 | 3 | 3 | 3 | 3 | 2.83 | 3 | 3 | 3 | 3 | 3 | 3 | 3.00 |

The Cohen's kappa value for the quality of responses under this category was calculated as 0.84 for ChatGPT-3.5 and 0.96 for ChatGPT-4. According to Landis and Koch's (1977) benchmark, these results indicate 'almost perfect' agreement on inter-raters with regard to the quality of these responses in terms of accuracy, clarity, conciseness, and potential bias.

Our next query on ChatGPT was about 'teacher technology integration', and sample excerpts from the responses of both versions of ChatGPT are shown in Figure 3.

Analysis of responses generated by ChatGPT for the third category of our query showed that the two versions produced quite different information with regard to the factors affecting teachers' technology integration. The responses by ChatGPT-3.5 were shallower and more generalized, while ChatGPT-4 provided more comprehensive and detailed information presented in clear categories.

RESEARCH THEME: Factors Affecting Teachers' Technology Integration

Model: GPT-3.5

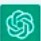 What are the factors that affect teachers' technology integration?

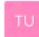 There are several factors that can affect teachers' technology integration, including:

1. Technological infrastructure: Access to reliable and up-to-date technological infrastructure, such as computers, software, and internet connectivity, is essential for effective technology integration. If the technological infrastructure is inadequate or unreliable, it can hinder teachers' ability to use technology in their teaching.

2. Technical support: Teachers require adequate technical support to help them troubleshoot technical problems and to provide assistance in using new technology tools and resources. Without adequate technical support, teachers may be hesitant to integrate technology into their teaching.

Model: GPT-4

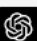 What are the factors that affect teachers' technology integration?

Several factors can affect teachers' technology integration in their classrooms, influencing their ability and willingness to incorporate digital tools and resources into their teaching practices. These factors can be grouped into individual, contextual, and technological categories:

1. Individual factors:

    a. Teacher attitudes and beliefs: Teachers' attitudes towards technology, their self-efficacy, and beliefs about the effectiveness of technology in enhancing teaching and learning play a significant role in their willingness to integrate technology.

    b. Prior experience and knowledge: Teachers who have previous experience using technology or have received training in technology integration are more likely to incorporate digital tools into their classrooms.

    c. Confidence and competence: Teachers' confidence and competence in using technology can influence their technology integration. Teachers who feel comfortable using digital tools are more likely to adopt them in their teaching practices.

2. Contextual factors:

    a. School culture and leadership: The school's culture, including the extent to which innovation is encouraged, and the support provided by school leaders, can significantly impact teachers' technology integration.

    b. Professional development opportunities: Access to ongoing training and professional development in technology integration can help teachers build their skills and confidence, increasing their likelihood of using technology effectively in their classrooms.

**Figure 3.** Sample excerpts from responses of ChatGPT-3.5 and ChatGPT-4 for teachers' technology integration (generated on 13 April 2023).

The responses generated by ChatGPT-3.5 regarding the factors that affect teachers' technology integration included technological infrastructure, technical support, professional development, attitudes and beliefs, pedagogical practices, time constraints, curriculum

and assessment. On the other hand, ChatGPT-4 provided more comprehensive answers, categorizing a total of 12 factors under 3 main categories: individual, contextual and technological factors. The category of individual factors comprised the sub-categories of teacher attitudes and beliefs, prior experience and knowledge, confidence and competence, while the category of contextual factors comprised the sub-categories of school culture and leadership, professional development opportunities, curriculum and assessment requirements, time constraints, and access to resources and support. The technological factors category included availability and accessibility, reliability and ease of use, compatibility with existing practices, and perceived usefulness sub-categories. In addition, ChatGPT-4 emphasized the importance of these factors for supporting schools and educational employees.

The average values of ratings based on the four-dimensional rating scale for the accuracy, clarity, conciseness, and potential bias in the information provided by ChatGPT-3.5, were 2.50, 2.50, 2.00, and 2.67, respectively, while it was 2.83, 2.67, 3, and 2.83 for ChatGPT-4 responses (see Table 3). These results indicate that both versions of ChatGPT were able to generate satisfactory information on identifying factors that affect teachers' technology integration. However, researchers clearly thought that ChatGPT-4 generated better responses, particularly in terms of accuracy and conciseness of information with lesser possibility of bias.

**Table 3.** The Rating Scores for the Factors Affecting Teachers' Technology Integration.

| | ChatGPT-3.5 | | | | | | | ChatGPT-4 | | | | | | |
|---|---|---|---|---|---|---|---|---|---|---|---|---|---|---|
| | R1 | R2 | R3 | R4 | R5 | R6 | $\bar{x}$ | R1 | R2 | R3 | R4 | R5 | R6 | $\bar{x}$ |
| Accuracy | 2 | 2 | 3 | 3 | 3 | 2 | 2.50 | 3 | 3 | 2 | 3 | 3 | 3 | 2.83 |
| Clarity | 3 | 2 | 3 | 3 | 2 | 2 | 2.50 | 3 | 3 | 2 | 2 | 3 | 3 | 2.67 |
| Conciseness | 2 | 2 | 2 | 2 | 2 | 2 | 2.00 | 3 | 3 | 3 | 3 | 3 | 3 | 3.00 |
| Bias Possibility | 2 | 3 | 3 | 2 | 3 | 3 | 2.67 | 3 | 2 | 3 | 3 | 3 | 3 | 2.83 |

Cohen's kappa value for the quality of responses under the category of factors affecting teachers' technology integration was calculated as 0.87 for ChatGPT-3.5 and 0.98 for ChatGPT-4. According to Landis and Koch's (1977) benchmark, these results indicate 'almost perfect' inter-rater agreement with regard to the quality of these responses with regard to the accuracy, clarity, conciseness, and bias possibility of information in terms of accuracy, clarity, conciseness, and potential bias.

Our final query addressed 'the impact of digital leadership on teachers' technology integration', and sample excerpts from the responses of both versions of ChatGPT are shown in Figure 4.

Regarding the effect of school principals' digital leadership on teachers' technology integration, both versions of ChatGPT emphasized the role of digital leaders in encouraging teachers to use new technologies and pointed to the significant impact of principals' digital leadership on teachers' technology integration. However, ChatGPT4 generated more comprehensive and categorical responses and summarized the effects of principals' digital leadership on teachers' technology integration under these categories: vision and strategy, school culture, professional development, collaboration and communication, data-driven decision making, encouragement, and recognition. ChatGPT-3.5 emphasized that the encouragement of school principals to enhance professional development opportunities was significant in enabling teachers' successful integration of technology into the classroom. However, in addition to this information, ChatGPT-4 also underlined the significance of building collaboration and cooperation between not only the principal and the teachers but also among teachers themselves and states that the principal should be a role model in this regard. ChatGPT-4 also finished its response stating that "*by prioritizing digital leadership, principals can help foster an innovative and technology-rich learning environment that*

*benefits both teachers and students*", which complies with the arguments in the literature (e.g., Agustina et al. 2020; Arham et al. 2023).

---

RESEARCH THEME: The Impact of Digital Leadership in Teachers' Technology Integration

Model: GPT-3.5

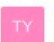 Does school principals' digital leadership affect teachers' technology integration?

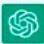 Yes, school principals' digital leadership can have a significant impact on teachers' technology integration. School principals play a key role in creating a culture of innovation and continuous improvement, where educators are encouraged to experiment with new technologies and approaches to teaching, and to share their experiences and insights with colleagues.

Effective digital leadership by school principals can provide a clear vision and direction for technology integration, and can help to establish a shared understanding of the role that technology plays in teaching and learning. School principals who are knowledgeable and skilled in technology integration can provide guidance and support to teachers, and can create a positive and supportive environment that encourages teachers to take risks and try new approaches.

Model: GPT-4

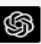 Does school principals' digital leadership affect teachers' technology integration?

Yes, school principals' digital leadership can significantly impact teachers' technology integration. As leaders in their schools, principals play a critical role in creating an environment that fosters innovation, supports technology adoption, and encourages effective use of digital tools in teaching practices. The influence of principals' digital leadership on teachers' technology integration can be seen in the following ways:

1. Vision and strategy: Principals who have a clear vision and strategy for integrating technology into their schools help provide direction and purpose for teachers. When teachers understand the goals and expectations related to technology use, they are more likely to embrace technology integration.
2. School culture: Principals who cultivate a positive school culture that encourages innovation, collaboration, and risk-taking can create an environment where teachers feel supported and motivated to experiment with new technologies and teaching approaches.

---

**Figure 4.** Sample excerpts from responses of ChatGPT-3.5 and ChatGPT-4 for the impact of digital leadership on teachers' technology integration (generated on 13 April 2023).

The average values of ratings based on the three-dimensional rating scale for the accuracy, clarity, conciseness, and potential bias in the information provided by ChatGPT-3.5 were 2.67, 2.67, 2.16, and 2.83, respectively, while it was 3, 2.83, 2,83, and 3 for ChatGPT-4 responses (see Table 4). These results indicate that both versions of ChatGPT were able to generate satisfactory information on the impact of digital leadership skills of school principals on teachers' technology integration. These results also indicate that ChatGPT-4 was considered to have generated better responses in terms of accuracy, clarity, and conciseness of information with a weaker potential for bias.

The values of Cohen's kappa for the quality of responses with regard to the impact of school principals' digital leadership on teachers' technology integration were calculated

as 0.86 for ChatGPT-3.5 and 0.97 for ChatGPT-4. According to Landis and Koch's (1977) benchmark, these results indicate 'almost perfect' inter-rater agreement with regard to the quality of these responses in terms of accuracy, clarity, conciseness, and potential bias.

**Table 4.** The Rating Scores for the Impact of Digital Leadership in Teachers' Technology Integration.

| | ChatGPT-3.5 | | | | | | | ChatGPT-4 | | | | | | |
|---|---|---|---|---|---|---|---|---|---|---|---|---|---|---|
| | R1 | R2 | R3 | R4 | R5 | R6 | $\bar{x}$ | R1 | R2 | R3 | R4 | R5 | R6 | $\bar{x}$ |
| Accuracy | 2 | 3 | 2 | 3 | 3 | 3 | 2.67 | 3 | 3 | 3 | 3 | 3 | 3 | 3.00 |
| Clarity | 3 | 3 | 2 | 2 | 3 | 3 | 2.67 | 3 | 3 | 3 | 2 | 3 | 3 | 2.83 |
| Conciseness | 2 | 2 | 2 | 2 | 3 | 2 | 2.16 | 3 | 3 | 2 | 3 | 3 | 3 | 2.83 |
| Bias Possibility | 2 | 3 | 3 | 3 | 3 | 3 | 2.83 | 3 | 3 | 3 | 3 | 3 | 3 | 3.00 |

## 4. Discussion

This study investigated digital school leadership and its role in teachers' technology integration via responses generated by the two versions (3.5 and 4) of ChatGPT. The study particularly tested the utility of ChatGPT as an alternative source of information that could support the work of researchers.

We observed that both versions of ChatGPT (3.5 and 4) generated a significant amount of accurate information to respond to our research questions. The responses generated for the definition of digital leadership were quite similar in both versions of ChatGPT. While defining digital leadership, ChatGPT-3.5 focused on the integration of technology into the teaching-learning process, whereas ChatGPT-4 emphasized its integration into school management processes as well. It is noteworthy that the definitions made by both versions of ChatGPT were sufficient and convincing, and at the same time, they were similar to the definitions cited in the literature. For instance, Zhong (2017) defined digital leadership as the use of school technologies, including digital tools and resources, to lead digital transformation, create a digital learning culture and provide professional development. Both versions of ChatGPT offered responses similar to this definition. In addition, as underlined by recent evidence, the necessity of leadership in promoting effective integration of technology into education (Al-Ruz and Khasawneh 2011; Keengwe et al. 2008) was also emphasized in the responses by ChatGPT.

As for the query on digital leadership skills, both versions of ChatGPT gave similar responses, describing these skills as competencies to integrate technology into the teaching and learning process. Both versions of ChatGPT emphasized digital leadership skills such as technological proficiency, communication, collaboration, change management, and data literacy. In addition to these skills, ChatGPT-4 articulated concepts such as digital transformation, the digital age, digital citizenship, and data-based decision making. It also generated more comprehensive and categorized responses regarding digital leadership skills. In addition, ChatGPT-4 emphasized the guidance and encouragement roles of school principals in performing successful digital leadership. Eberl and Drews (2021) stated that one of the determinants of effective digital leadership is cooperation and communication in addition to building technology-enhanced decision processes that would enable agile and fast solutions. Tigre et al. (2023) stated that successful digital leadership encompassed effective communication, guidance, transparency, trust, agility, collaboration, innovation, empowerment, and adaptability. Similarly, Bolte et al. (2018) emphasized that communication is an important skill for digital leaders, while Kane et al. (2019) stated that digital leaders should have an innovative mindset and act collaboratively. Karakose et al. (2021) reported that digital leadership basically requires technological skills, managerial skills, and personal skills. They also underlined that digital leaders should have many other capabilities, such as digital literacy, change management, cooperation, vision building, trust, participative decision making, and communication.

With regard to teachers' technology integration, we observed that ChatGPT-3.5 gave more superficial and cursory responses, whereas ChatGPT-4 was able to generate more comprehensive responses. In addition, ChatGPT-4 responded by categorizing a total of 12 skills under the headings of individual, contextual and technological factors, including the responses of ChatGPT-3.5. The categorical presentation of information made these responses clearer and more concise compared to those generated by ChatGPT-3.5. Considering these responses with reference to the relevant literature, many studies have concluded that teachers' attitudes and beliefs directly affect their integration of technology into the classroom (Alghamdi and Prestridge 2015; Christensen 2002; Ertmer et al. 2012; Kim et al. 2013; Liu 2011). According to Karakose et al. (2021), as digital leaders, school principals promote a digital learning culture in schools by creating opportunities for professional development and enabling digital transformation with technology-based interventions in schools. Similarly, Shuldman (2004) underlined that the lack of time and resources is a significant obstacle to teachers' integration of technology into the classroom. All these aspects with regard to supporting teachers' technology integration into the teaching-learning processes were mentioned by both versions of ChatGPT, but ChatGPT-4 offered broader information with a better synthesis of these factors.

Regarding the impact of school principals' digital leadership on teachers' technology integration, both versions of ChatGPT focused on the role of digital leaders in encouraging teachers to use new technologies. ChatGPT-4 articulated more categorical responses and classified these effects under the headings of vision and strategy, school culture, professional development, collaboration and communication, and data-based decision making. McLeod (2015) stated that successful schools had a strong vision of empowering students to use digital technologies for the benefit of their learning. Similarly, Eberl and Drews (2021) stated that digital leadership had a significant role in developing a culture that encouraged employees to act with curiosity, think out of the box, and constantly expand their knowledge so that they could transform both themselves and the organization they work for. Both versions of ChatGPT emphasized the roles of school principals and underlined that creating opportunities for professional development had a significant impact on teachers' integration of technology into education. Scholars also stated that school principals had a positive effect on teachers' willingness for and capability to integrate technology into the classroom (Agustina et al. 2020; Chang 2012). In addition, Kane et al. (2019) emphasized that developing network-based leadership rather than establishing hierarchies facilitates faster decision making and functioning capacity of organizations, which is key to the digital transformation expected from modern organizations in the face of rapid technological advancements. ChatGPT-4 also pointed to this aspect of digital leadership by articulating that digital leaders can create networks of experience by collaborating.

## 5. Conclusions

The current study conducted a comparative assessment of the two versions of an AI-based LLM, ChatGPT-3.5 and ChatGPT-4, with regard to their capacity to provide accurate, clear, concise information with a weaker possibility of bias, which are essential components of scientific knowledge creation. The results of the queries conducted on both versions of ChatGPT with regard to digital school leadership and its relation to teachers' technology integration showed that these AI-based tools are promising in terms of helping researchers save time and reach synthesized information on a given topic just in seconds. However, we should also caution that, in its current form, these tools are still in their infancy, and there is a long way to go before they become fully capable of supporting scientific work. It is also important to remember that information provided by ChatGPT is currently limited to their training corpus, and although they are trained on a large corpus of texts, including academic papers (OpenAI 2023a), the information currently provided is inevitably biased by the content of their training corpus.

As stated by Ollivier et al. (2023), AI-based technologies are like a modern Pandora's box that is already open with many opportunities and threats, and the same holds true for

the potential use of LLMs like ChatGPT in the scientific research process. Although several concerns and potential use cases have already been identified in the literature, there is still a need for more comprehensive studies to ensure the responsible, safe and ethical integration of these technologies into the process of scientific knowledge creation because burying our heads into the sand with complete ignorance would be of no use since that box has already been opened. Therefore, we strongly recommend that future research should continue to address the opportunities and threats of ChatGPT with scientific methods and identify ways of maximizing its benefits while reducing possible risks, such as generating unethical or misleading results, as well as overcoming accountability or authorship concerns. Multidisciplinary research conducted with the participation of experts from a large scope of fields could particularly be useful to address the issue from multiple perspectives and yield well-rounded results. In addition, studies addressing the perspectives of scientists on the possible benefits and threats of using ChatGPT, conferring on their ideas regarding the effective or fraudulent use of these technologies for scientific work through qualitative research designs, could provide deeper insights in this regard. It should also be noted that AI-based technologies continue to develop, and their newer versions keep being released, so the relevant literature warrants updating in parallel with these developments. In addition, the current study particularly aimed to conduct a comparative analysis between the responses generated by the two versions of ChatGPT over a single set of queries. However, future research could also conduct a similar study using responses generated from a cycle of multiple queries so as to conduct temporal comparisons and identify any differences in the quality of the responses generated over different time intervals.

Despite its contribution to the debates over the potential use of ChatGPT for scientific work through comparative assessment of ChatGPT-3.5 and ChatGPT 4 responses, the current study bears some limitations. One major limitation is that the assessments of ChatGPT content were based on the subjective evaluations of researchers to a large extent, although their evaluations relied on an extensive literature review of digital school leadership and teacher technology integration, and a variety of evaluation methods were used to obtain the results presented in the study. Another limitation is that queries in the current study were made in only English, so the results should not be generalized to other languages. Considering that ChatGPT could generate responses in several languages, a similar study design could be realized in other languages to identify divergent and convergent results. Finally, although the current study yielded optimistic results regarding the potential contribution of ChatGPT to identifying, accumulating, and categorizing a large amount of information in a matter of minutes, it is crucial to remember that we should continue critically evaluating the accuracy and utility of information generated by these LLMs as they still lack scientific reasoning and critical thinking, and have the potential to generate infodemics or artificial hallucinations (Alkaissi and McFarlane 2023; Ollivier et al. 2023) although not observed in the current analysis. This difference in results could have resulted from the topics under investigation or could be specific to research fields such as medicine, economics, physics, or law. Further investigations could increase our understanding with regard to this topic and would contribute greatly to ChatGPT literature.

**Author Contributions:** Conceptualization, T.K.; methodology, T.K. and T.T.; formal analysis, T.K., T.T., R.Y., M.D., H.P. and T.Y.O.; data curation, T.K., T.Y.O. and R.Y.; writing—original draft preparation, T.K., T.T. and H.P.; writing, T.K., T.T., R.Y., M.D., H.P. and T.Y.O.; review and editing, T.K., T.T., M.D., H.P., T.Y.O. and R.Y.; supervision, T.K. All authors have read and agreed to the published version of the manuscript.

**Funding:** This research received no external funding.

**Institutional Review Board Statement:** Not applicable.

**Informed Consent Statement:** Not applicable.

**Data Availability Statement:** No identifying information was collected or included. All the data used in the results section of this research was accessed through ChatGPT-3.5 and ChatGPT 4.

**Acknowledgments:** The authors acknowledge the contribution of both versions of ChatGPT and thank the working team of OpenAI for supporting this work through the development of this generative AI-based language model.

**Conflicts of Interest:** The authors declare no conflict of interest.

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
