# Peer review of "A Conversation with ChatGPT about Digital Leadership and Technology Integration: Comparative Analysis Based on Human–AI Collaboration"

_admsci, doi:10.3390/admsci13070157_

Round 1
Reviewer 1 Report
The article itself is fascinating, not only because we are living in an age of AI but also because of the originality and depth that it provides to the reader. We agree that nowadays, Pandora's Box is already open. This article discusses precisely the importance of analysing the content provided by an LLM language like CHATGPT. This is what we need to discuss today, not tomorrow, but today, to learn more about these systems and to know how to use them for our benefit.
Author Response
We’d like to thank you for this positive and encouraging feedback.
Reviewer 2 Report
Frankly speaking, the paper is not so relevant to the scope of Administrative Sciences in my understanding. I want the journal editor to make a final decision, as that seems to be beyond the responsibility of a reviewer, but it may be better to be transferred to a more relevant MDPI journal. It is more appropriate to be published in information-technology-oriented journals. If the evaluation of the different versions of ChatGPT is suggestive of theoretical discussion regarding digital leadership and technology integration, the paper should be published in AS. But I am afraid this is not the case.
Even if the evaluation of ChatGPT responses (regarding the subject matter seemingly relevant to the journal scope) is acceptable in the journal, I still have a concern about the analytical perspective of the study.
For example, more detailed comparative investigations on the results not only by the four categories but by three dimensions of rating scales and potential bias. Then you can identify the gap between the two versions of ChatGPT more clearly from multiple perspectives. I am afraid the current version of the discussion section lacks this systematic and persuasive argument.
For the other, you should elaborate on the literature generally on the development of ChatGPT, particularly the researchers' evaluation of the improvement by version 4 compared with version 3.5. The authors will be able to show how their results are similar to and/or different from the literature.
I don't have the expertise and experience to comment on the present study's evaluation of the applicability of ChatGPT to this subject matter.
ll.88-89: It is very much confusing to elaborate parallelly technology advancements in education and developments in AI and machine learning, especially because the former is the subject matter while the latter is the method to analyze the subject matter (moreover, we should note that the present study analyzed the analysis of the subject matter by the method, or you have a meta-structure here).
ll.129-134: We may tell so only when ChatGPT finds new perspectives that have not been argued by human researchers. At the stage of evaluating how ChatGPT's answers are similar to the literature (with a precondition that it is inferior to humans), we cannot argue this way.
Ll.333-356: These arguments on ChatGPT should have been made before showing the result, especially because they are not discussions on the results.
Author Response
|
REVIEWER 2 |
|
|
Reviewer’s Comments |
Authors’ Response |
|
more detailed comparative investigations on the results not only by the four categories but by three dimensions of rating scales and potential bias. Then you can identify the gap between the two versions of ChatGPT more clearly from multiple perspectives. I am afraid the current version of the discussion section lacks this systematic and persuasive argument.
|
Thank you for your feedback. In light of your comments, we elaborated comparisons between ChatGPT-3.5 and GPT-4’s responses in the results section through citing illustrative sections from the raw responses. |
|
you should elaborate on the literature generally on the development of ChatGPT, particularly the researchers' evaluation of the improvement by version 4 compared with version 3.5. The authors will be able to show how their results are similar to and/or different from the literature
|
Thank you for this feedback. We re-organized the introduction section upon your comment and started this section elaborating on ChatGPT, and then presented theoretical information about digital leadership and teachers’ technology integration.
|
|
88-89: It is very much confusing to elaborate parallelly technology advancements in education and developments in AI and machine learning, especially because the former is the subject matter while the latter is the method to analyze the subject matter (moreover, we should note that the present study analyzed the analysis of the subject matter by the method, or you have a meta-structure here).
|
Agreeing on your comment, we removed this section completely and added the explanation about machine learning and AI in the upper sections of introduction, before we start writing about ChatGPT. |
|
129-134: We may tell so only when ChatGPT finds new perspectives that have not been argued by human researchers. At the stage of evaluating how ChatGPT's answers are similar to the literature (with a precondition that it is inferior to humans), we cannot argue this way
|
Thank you for your feedback. We removed this explanation from this section. |
|
333-356: These arguments on ChatGPT should have been made before showing the result, especially because they are not discussions on the results.
|
Thank you for the comment. We removed these two paragraphs into the introduction section, right after the comparison of ChatGPT-3.5 and GPT-4. |
Reviewer 3 Report
The paper provides an assessment of ChatGPT large language model in supporting scientific research. The authors conducted several queries using two versions of ChatGPT and later evaluated the generated responses in terms of accuracy, clarity, conciseness, and bias. Their results indicate that ChatGPT shows promise in providing synthesized information on research topics, but it is still in its early stage and requires further development.
The authors mention that the assessments of ChatGPT's content relied largely on subjective evaluations by researchers. The subjective nature of the evaluations may introduce some level of bias. It would be beneficial to add the characteristics of the diversity of researchers assessing the ChatGPT answers. Moreover, there should be in-depth discussion of the potential sources of bias and its implications for the reliability of the information provided by ChatGPT.
While the paper mentions the potential of LLMs to generate infodemics or hallucinations, no specific analysis regarding these concerns are presented. As the number of queries were very limited (4 queries presented in the paper), the analysis is quite shallow and the results can be misleading. Therefore, much more details on the specific methodology should be explained, such as the selection criteria for queries and the process of analyzing the responses, to provide clarity and help assess the reliability and validity of the findings.
The authors mention the usage of a three-dimensional rating scale. But in fact it is a scale that could be summarized as: 1 - negative, 2 - partly fine, 3 - positive (in the aspects of being accurate, clear, concise, or biased). This to me seems to be just one dimension with 3 values in this dimention, not three-dimensional. If the authors assessed separately being 1) accurate, 2) clear, 3) concise, 4) biased, they could have four-dimentional rating. More information is needed also on the specific criteria and inter-rater agreement statistics.
I have not spotted any language issues.
Author Response
|
REVIEWER 3 |
|
|
Reviewer’s Comments |
Authors’ Response |
|
The authors mention that the assessments of ChatGPT's content relied largely on subjective evaluations by researchers. The subjective nature of the evaluations may introduce some level of bias. It would be beneficial to add the characteristics of the diversity of researchers assessing the ChatGPT answers. Moreover, there should be in-depth discussion of the potential sources of bias and its implications for the reliability of the information provided by ChatGPT.
|
Thank your for this informative feedback. Upon your comment, we added a paragraph at the end of the materials and methods section, and gave detailed information about the researchers participated in the analysis. |
|
While the paper mentions the potential of LLMs to generate infodemics or hallucinations, no specific analysis regarding these concerns are presented. As the number of queries were very limited (4 queries presented in the paper), the analysis is quite shallow and the results can be misleading. Therefore, much more details on the specific methodology should be explained, such as the selection criteria for queries and the process of analyzing the responses, to provide clarity and help assess the reliability and validity of the findings.
|
As you have also underlined, data for the current analysis was specific to the topic of digital leadership and teachers’ technology integration and our results did not illustrate any sign of hallucinations with regard to our queries. However, we also wanted to make this warning as generating infodemics and hallucinations was mentioned by some previous research from other fields of study such as medicine or economics. This result could be related to topic under investigation or to the characteristics of the research field. We considered that this could be an interesting point to be investigated in future studies, and added this as a future research suggestion into the relevant section.
|
|
The authors mention the usage of a three-dimensional rating scale. But in fact it is a scale that could be summarized as: 1 - negative, 2 - partly fine, 3 - positive (in the aspects of being accurate, clear, concise, or biased). This to me seems to be just one dimension with 3 values in this dimension, not three-dimensional. If the authors assessed separately being 1) accurate, 2) clear, 3) concise, 4) biased, they could have four-dimensional rating. More information is needed also on the specific criteria and inter-rater agreement statistics.
|
Thank you very much for this feedback. We made a mistake while writing about the scale and you are totally right that it was a four- dimensional scale with three rating points. We made the necessary corrections in the relevant section. |
Round 2
Reviewer 2 Report
I appreciate your effort in revising the paper, especially about more detailed descriptions in the result section. But I am afraid I found some critical issues because of these elaborations, and you are responsible for justification. Otherwise, the paper is not qualified for publication, in my understanding. Hope that you will make another round of revision successfully.
One more thing is, even though I wanted you to reply to my general comment on the relevance of the present study to Administrative Sciences, you may not be responsible, so I will not ask you again.
Let me make some more specific comments below.
ll.293-298: Is "more positive" better than "more neutral" for rating? If this is just an interpretation and does not affect ratings, there is no substantive problem. But if it is one of the criteria for evaluation, it may become a critical issue for the other categories as well. I have the same concern about the impact of digital 394 leadership on teachers' technology integration (ll. 409-412).
Much more critically, I cannot find the phrases cited as parts of the answers by ChatGPT in the figures. For example, "By embracing digital leadership" and "Overall, digital 297 leadership is about" are not found in Figure 1. Check the consistency between the descriptions in the body and the other figures. If those phrases were not included in the responses by ChatGPT, where did they come from? I see Figure 1 shows the "excerpts", but something cited in the main body should be included in the excerpts. I cannot help suspecting how your excerpts summarized the full responses appropriately. Due to this suspicion, I now want you to submit the full responses as an appendix or at least supplementary information for the reviewers and explain the process of summarizing the full responses. In fact, I do not request you this sort of thing if you clearly mention that "although the following are not included in the excerpts…" with reasonable justification. But please understand as a reviewer, I have to confirm if the authors are conscious of the potential problem.
Author Response
|
REVIEWER 2 |
|
|
Reviewer’s Comments |
Authors’ Response |
|
ll.293-298: Is "more positive" better than "more neutral" for rating? If this is just an interpretation and does not affect ratings, there is no substantive problem. But if it is one of the criteria for evaluation, it may become a critical issue for the other categories as well. I have the same concern about the impact of digital 394 leadership on teachers' technology integration (ll. 409-412). |
In fact, with the sentences/phrases you underlined here, we wanted to underline that some answers were more inclined to be similar to scholarly arguments in the literature. For instance, the literature on digital leadership stays positive about adopting digital leadership in schools in the current era and this stance was evident in GPT-4’s answers. However, being positive or neutral was not our criteria on itself, and was not among the four criteria we mentioned in the manuscript. However, you notification was useful to avoid such misunderstandings, and we slightly revised these sections you underlined.
|
|
Much more critically, I cannot find the phrases cited as parts of the answers by ChatGPT in the figures. For example, "By embracing digital leadership" and "Overall, digital 297 leadership is about" are not found in Figure 1. Check the consistency between the descriptions in the body and the other figures. If those phrases were not included in the responses by ChatGPT, where did they come from? I see Figure 1 shows the "excerpts", but something cited in the main body should be included in the excerpts. I cannot help suspecting how your excerpts summarized the full responses appropriately. Due to this suspicion, I now want you to submit the full responses as an appendix or at least supplementary information for the reviewers and explain the process of summarizing the full responses. In fact, I do not request you this sort of thing if you clearly mention that "although the following are not included in the excerpts…" with reasonable justification. But please understand as a reviewer, I have to confirm if the authors are conscious of the potential problem |
While uploading our manuscript to the journal website, we actually enclosed a supplementary file which includes the full responses generated by the both versions of ChatGPT during our queries, considering that the editor or you as reviewers would like to see the full responses. Including all the responses in the manuscript makes it unnecessarily lengthy, and looks as if the raw data is presented without any analysis. That’s why we preferred to include the excerpts in Figures to illustrate the queries. However, if requested, the tables in the supplementary file could well be inserted into the appendix.
If you would like to see the full responses and currently cannot access to the supplementary file on the system, could you please contact the editor asking for it. We also wrote a massage to the editor, asking if it was possible to share the supplementary file with the reviewers upon request, and hope she would help with that. |
Reviewer 3 Report
The paper has been slightly improved, however, the presented research in my opinion is biased.
The authors only provide the average values of the ratings of 6 participants. There is no information about the statistics of these ratings (min/max/standard deviation).
Moreover, only a single answer to a question is obtained.
As ChatGPT can generate multiple contradictory answers, for the research to be valid, I suggest querying the model with the same prompts multiple times and providing the assessment for several answers, and analyze also the distribution for these answers within single question scope.
Otherwise, the provided research is just biased and not reliable.
Therefore, other researchers might easily have quite opposite findings.
l. 409: ChatgPT-4 => ChatGPT-4
Author Response
|
REVIEWER 3 |
|
|
Reviewer’s Comments |
Authors’ Response |
|
The authors only provide the average values of the ratings of 6 participants. There is no information about the statistics of these ratings (min/max/standard deviation). |
The ratings in the present study were made by six researchers under the four categories (e. i. the accuracy, clarity, conciseness, potential bias). Therefore, we only calculated the averages of the rating points assigned by each researcher for each category of evaluation. In other words, the process did not require statistical analysis. However, upon your feedback, we included the rating points for each category in the form of Tables to make the process more transparent.
|
|
Moreover, only a single answer to a question is obtained. As ChatGPT can generate multiple contradictory answers, for the research to be valid, I suggest querying the model with the same prompts multiple times and providing the assessment for several answers, and analyze also the distribution for these answers within single question scope. Otherwise, the provided research is just biased and not reliable. |
Thank you for this feedback. As the authors of this manuscript, we believe we took several steps to minimize the bias in our research through transparently describing every step we too during the research and we also notified the limitations as well. Therefore, we believe the readers of the manuscript are well informed about the purpose, the strengths and the limitations of the study, which we believe supports the rigor of our study.
In the current study, our purpose was not to test the differences of responses yielded over multiple times of query but we aimed to conduct a comparison between the responses generated by the two versions of ChatGPT. Therefore, we believe conducting queries simultaneously (i. e., at the same time on two computers) on both versions was more significant than having multiple queries. In the literature, there are some studies who conducted multiple queries, but to our knowledge, their purpose includes making comparisons between the responses yielded at different times by the same version of ChatGPT (namely, Chat-GPT 3.5). However, as the authors, we also considered that the suggestions you made here could be the subject of another research, and could be a good idea for future research. Therefore, we included this suggestion in the implications for future research section.
|
|
l. 409: ChatgPT-4 => ChatGPT-4 |
Thank you very much for the notification. We corrected the typing mistake. |
Round 3
Reviewer 2 Report
I am terribly sorry that I did not notice you shared the full responses as supplementary material, due to my careless misunderstanding. I acknowledge my faults and I agree with your decision that you did not include the full responses from ChatGPT either in the main body or as an appendix. But in the manuscript, can you provide the link to the full responses as the supplementary file?
Thank you for your revision to my latest comment.
Reviewer 3 Report
The authors answered my comments. I do not have further suggestions.